# Metallothionein Family Proteins as Regulators of Zinc Ions Synergistically Enhance the Anticancer Effect of Cannabidiol in Human Colorectal Cancer Cells

**DOI:** 10.3390/ijms242316621

**Published:** 2023-11-22

**Authors:** In-Seo Kwon, Yu-Na Hwang, Ju-Hee Park, Han-Heom Na, Tae-Hyung Kwon, Jin-Sung Park, Keun-Cheol Kim

**Affiliations:** 1Department of Biological Sciences, College of Natural Sciences, Kangwon National University, Chuncheon 24341, Kangwon, Republic of Korea; kwoin087@kangwon.ac.kr (I.-S.K.); qha1259@kangwon.ac.kr (Y.-N.H.); pjhee@kangwon.ac.kr (J.-H.P.); hanhum01@kangwon.ac.kr (H.-H.N.); 2Kangwon Center for System Imaging, Chuncheon 24341, Kangwon, Republic of Korea; 3Chuncheon Bioindustry Foundation, Chuncheon 24232, Kangwon, Republic of Korea; team0218@cbf.or.kr; 4Korean Pharmacopuncture Institute, Seoul 07525, Republic of Korea; omd9339@mapi.or.kr

**Keywords:** cannabidiol, colorectal cancers, cell death, metallothionein, zinc ion

## Abstract

Cannabidiol (CBD) is a chemical obtained from *Cannabis sativa*; it has therapeutic effects on anxiety and cognition and anti-inflammatory properties. Although pharmacological applications of CBD in many types of tumors have recently been reported, the mechanism of action of CBD is not yet fully understood. In this study, we perform an mRNA-seq analysis to identify the target genes of CBD after determining the cytotoxic concentrations of CBD using an MTT assay. CBD treatment regulated the expression of genes related to DNA repair and cell division, with metallothionein (MT) family genes being identified as having highly increased expression levels induced by CBD. It was also found that the expression levels of MT family genes were decreased in colorectal cancer tissues compared to those in normal tissues, indicating that the downregulation of MT family genes might be highly associated with colorectal tumor progression. A qPCR experiment revealed that the expression levels of MT family genes were increased by CBD. Moreover, MT family genes were regulated by CBD or crude extract but not by other cannabinoids, suggesting that the expression of MT family genes was specifically induced by CBD. A synergistic effect between CBD and MT gene transfection or zinc ion treatment was found. In conclusion, MT family genes as novel target genes could synergistically increase the anticancer activity of CBD by regulating the zinc ions in human colorectal cancer cells.

## 1. Introduction

Colorectal cancer (CRC) is the third most frequently diagnosed cancer. It ranks second in cancer mortality statistics globally [1,2]. The standard approach to CRC treatment starts with surgical removal, followed by treatments such as radiation therapy and chemotherapy [3]. 5-Fluorouracil (5-FU), an anti-metabolite drug that works by inhibiting nucleotide synthetic enzyme thymidylate synthase (TS), has been introduced as a chemotherapy agent for patients with progressed or recurrent CRC after surgery [4]. However, 5-FU chemotherapy only has therapeutic efficacy for 10–15% of CRC patients [5]. Another study has suggested that EGFR inhibitors, such as cetuximab, can be applied as a targeted therapy for CRC [6]. EGFR inhibitors can interfere with cell cycle progression in the G1 phase by increasing the expression of p27^kip1^, and they can show anti-angiogenic activity through the inhibition of vascular endothelial growth factor (VEGF) [7]. However, the intrinsic and acquired resistance of EGFR inhibitors can be caused by various mechanisms, such as an EGFR mutation and an activated AKT/mTOR pathway [8]. Although novel chemotherapeutic approaches have been developed in order to increase the overall survival rate of cancer patients, there are still hurdles, such as side effects, personal variation, and acquired drug resistance [9].

The anticancer properties of phytocannabinoids extracted from *Cannabis sativa* (a hemp species) against many types of tumors have been studied [10]. The first study on cannabinoid activity was published in 1975, reporting the anticancer effects of cannabinoids against Lewis lung adenocarcinoma both in vitro and in vivo [11]. Treatment with Δ^9^-tetrahydrocannabinol (THC) and cannabinol (CBN) faces some legal issues because these compounds have psychoactive effects for utilization as anticancer drugs [12]. However, cannabidiol (CBD) and cannabigerol (CBG) do not have psychoactive effects. Thus, studies have been focused on CBD and CBG due to their potential use for cancer treatment [13]. CBD has antiproliferation and proapoptotic activities against breast cancer cells, including MDA-MB-231 and MCF-7, but not against MCF-10A normal cells [14]. CBD can increase the expression levels of COX-2-dependent prostaglandins (PGs) and PPAR-γ-dependent cell death in A549 xenograft nude mice [15]. CBD can mediate the cytotoxicity of human primary GBM stem-like cells (hGSCs) by promoting the DNA binding of NF-κB subunit RELA and simultaneously prevent RELA phosphorylation, suggesting that NF-κB can be converted into a tumor suppressor via non-psychotropic CBD [16]. Moreover, CBD can significantly reduce CRC invasion and metastasis by inhibiting the Wnt/β-catenin signaling pathway and the expression levels of β-catenin target genes, such as APC and CK1 [17]. These previous studies suggest that CBD could serve as a potential anticancer drug candidate that is highly effective against tumors [18].

Interestingly, CBD could synergistically increase the cytotoxicity of other anticancer drugs [19]. It has been shown that the encapsulation of CBD at a sub-optimal concentration (cell death < 10%) can extend antiproliferative activity for at least 10 days when CBD is combined with paclitaxel or doxorubicin in conventional breast cancer chemotherapy [20]. Consistently, CBD can potentiate the effects of common chemotherapy drugs, including cisplatin, 5-FU, and paclitaxel, in head and neck squamous cell carcinoma, achieved via CBD-induced cytotoxicity by decreasing the expression of genes involved in DNA repair [21]. In addition, CBD can potentially solve the problem of drug resistance that might occur in cancer treatment [22]. CBD shows strong cytotoxicity and tumorigenicity inhibitory effects in cisplatin-resistant human lung cancer cells by binding to the TRPV2 receptor, leading to the production of ROS and the inhibition of cell stemness [23]. CBD can inhibit FOXM1 and sensitize brain cancer cells to the DNA-damaging agent TMZ by downregulating the expression of RAD51, which encodes an important DNA damage repair protein [24]. CBD can suppress tumor growth both in vitro and in vivo by reducing phospho-NOS3 and SOD2 levels while inducing subsequent autophagy in oxaliplatin-resistant CRC cells [25]. ATP-binding cassette (ABC) transporters are responsible for releasing various drugs from cells, reducing intracellular drug concentrations and inducing drug resistance [26]. In ovarian cancer cells overexpressing ABCC1, CBD treatment can increase the intracellular accumulation of two ABCC1 substrates, Fluo3 and vincristine [27]. These previous papers suggest that the synergistic effect of CBD can maximize the effects of anticancer drugs, reduce their amounts, suppress drug resistance, and increase the death rates of cancer cells.

Given that the efficacy of chemotherapy depends on the tumor microenvironment, the endocannabinoid system (ECS), a major target of cannabinoids, has received much attention, as it involves diverse pathways in cancer biology [28]. CBD can act as an antagonist of CB_1_ and CB_2_ receptors in the ECS [29]. The core effect of the antagonism of CB_1_ receptors is a reduction in the binding affinity of THC and any of its related isomers [30]. However, CBD has agonistic effects on TRPV1 and TRPV2, indicating effects of CBD on seizure, inflammation, cancer, pain, acne, and vasorelaxation [31]. The anticancer effects of CBD are mainly mediated via its interaction with the ECS, resulting in the alleviation of pain and the promotion of immune regulation [32].

Although CBD has promising applications in cancer treatment, the way in which CBD cooperates with target proteins in molecular networks remains unclear. Thus, this study aimed to identify the target genes of CBD using an mRNA-seq analysis and to clarify the novel molecular mechanisms of CBD involving target genes.

## 2. Results

### 2.1. CBD Treatment Induces Growth Inhibition and Cell Death in Colorectal Cancer Cells

An MTT assay was performed to investigate the effect of CBD on the growth of SW480 and LoVo cells (Figure 1A). CBD at concentrations over 20 μM inhibited the growth of the two colorectal cancer cell lines, with IC_50_ values of 12.86 ± 0.527 μM and 23.03 ± 4.09 μM for the SW480 and LoVo cells, respectively. An FACS analysis showed that 20 μM CBD induced cell death by increasing the sub-G1 phase in a time-dependent manner (Figure 1B). The cell cycle distribution showed a slight increase in the G1 phase at 12 h after CBD treatment. The dead cell population was then increased at 24 h after CBD treatment. Annexin V-positive cells were also increased by 20 μM CBD (Figure 1C). Compared to the control, the percentage of Annexin V-positive cells increased by CBD was 22.0% for the SW480 cells and 24.91% for the LoVo cells. Therefore, CBD treatment could induce the growth inhibition and death of colorectal cancer cells.

### 2.2. mRNA-seq Analysis Shows Regulation of Multiple Genes by CBD Treatment

We performed an mRNA-seq analysis to identify differentially expressed genes after treatment with CBD. CBD regulated 1279 protein-coding genes in LoVo cells, with 323 upregulated genes and 956 downregulated genes after 24 h (Figure 2A). A GO term analysis suggested that CBD primarily regulated the expression of genes related to DNA replication or cell division, with most genes being downregulated by CBD treatment (Figure 2B). Interestingly, MT family genes, including MT1F, MT1G, MT1H, MT1M, MT1X, and MT2A, were dramatically upregulated by CBD treatment (Figure 2C). Among them, MT1G showed the highest expression change of up to 20-fold. The expression levels of most MT genes were increased more than 10-fold by CBD treatment. TCGA data suggested that most MT genes were downregulated in primary colorectal tumors compared to those in normal tissues (Figure 2D). A multiple gene analysis showed that the expression levels of MT1 isoforms and MT2A were lower in metastatic tissues and colorectal tumor tissues than in normal colorectal tissue using gene chip-based data (Figure 2E). These results suggest that CBD can regulate the expression of various genes and enhance the expression of MT family genes.

### 2.3. CBD Treatment Increases Expression of MT Family Genes

We performed a qRT-PCR analysis to investigate the expression of MT family genes after CBD treatment at different time periods. The expression levels of most MT genes were distinctly increased in the two colorectal cancer cell lines after 4 h of treatment and maintained increased expression for 24 h (Figure 3A). We also performed a Western blot analysis to investigate protein levels (Figure 3B and Appendix A). As a result, the MT and MT2A protein expression levels showed a consistently increased pattern depending on treatment time. An immunostaining analysis showed that the MT and MT2A expression levels in the cytoplasmic region of the colorectal cancer cells were increased by CBD treatment (Figure 4). These results suggest that CBD treatment is highly associated with the expression of MT family genes.

### 2.4. Expression Levels of MT Family Genes Are Specifically Regulated by CBD

To investigate whether other cannabinoids could regulate the expression of MT family genes, we treated colorectal cancer cells with several cannabinoids and investigated the expression of MT family genes (Figure 5A). qRT-PCR showed that the expression levels of MT family genes were increased by treatment with crude extract or CBD. However, no obvious increase in their expression was seen after treatment with Δ^9^-THC, CBG, or CBC. We also examined the levels of MT family proteins using a Western blot analysis (Figure 5B and Appendix A). MT and MT2A proteins were increased after treatment with crude extract or CBD. However, they showed no change after THC treatment. These results indicate that MT family gene expression is tightly regulated in the presence of CBD.

### 2.5. MT Overexpression Increases Cell Death Caused by CBD

We introduced overexpression vectors of the MT gene family to examine the functional association of MT protein expression with the anticancer effect of CBD. The population of Annexin V-positive cells was increased as much as 13.12% by MT1G overexpression compared to that in the control group (2.40% in the control and 3.03% in the vector control group) (Figure 6A). In addition, the dead cell population was increased as much as 41.42% upon MT1G overexpression with CBD treatment. We also obtained similar data after MT2A overexpression (Figure 6B). The dead cell population was increased as much as 34.30% upon MT2A overexpression with CBD treatment. A Western blot analysis showed the expression of FLAG-tagged-MT genes after transient transfection (inset of Figure 6B). Taken together, these results suggest that MT overexpression can synergistically increase the anticancer effect of CBD.

### 2.6. Cotreatment of CBD with Zinc Ions Enhances Dead Cell Population

MT family proteins play the main role in regulating the concentration of cytoplasmic zinc ions. Their expression levels are also regulated by zinc ions [33,34]. The expression levels of MT proteins in the two colorectal cancer cell lines were increased by zinc ion treatment according to a Western blot analysis, indicating that MT gene expression was regulated by CBD and zinc ions (Figure 7A and Appendix A). To examine the synergistic effect of CBD and zinc ions on anticancer activity, we performed an MTT assay using sublethal concentrations of CBD and zinc ions (Figure 7B). Although CBD at 5 μM did not show a growth inhibitory effect on colorectal cancer cells, combined treatment with CBD and 100 μM zinc ions significantly reduced cell viability compared to a single treatment. This synergistic effect showed a similar pattern in the cell cycle analysis experiment (Figure 7C). Dead cells were increased after combination treatment with CBD and zinc ions, suggesting that combinatory treatment with CBD and zinc ions could increase the cell death ratio. The dead cells were analyzed after Annexin V/PI staining (Figure 7D). The dead cell population of SW480 cells was increased to 72.4% after combination treatment with CBD and zinc ions, whereas it was 4.50% after treatment with CBD and 52.9% after treatment with zinc ions. For LoVo cells, the dead cell population was 18.6%. These results suggest that cotreatment with CBD and zinc ions can enhance the dead cell population of colorectal cancer cells.

## 3. Discussion

Cannabinoids can interact with their receptors or proteins via cellular pathways, thus affecting the development/progression of diseases, including cancer [35]. Our previous study suggested that CBD shows anticancer activity by inducing PPAR-γ-dependent cytoplasmic vesicles in human lung cancer cells and colorectal cancer cells [36,37]. This biological activity was also observed in colorectal cancer cells, indicating the anticancer effects of cannabinoids on various cancer types. CBD has been considered a safe drug, and CBD activity is strongly related to its dosage [38]. In vitro studies and animal experiments pointed to a concentration-dependent anticancer effect in micromolar ranges, although they were influenced by the nature of the cancer cells and the test conditions [39]. In addition, it has been reported that CBD might interact with a wide variety of molecular targets to exert its pharmacological effects [40]. Although many studies have shown promising anticancer activity of CBD, the profiles of target genes and multiple regulatory mechanisms mediating the anticancer activity of CBD remain unclear.

In this study, we first determined that 20 μM was a cytotoxic concentration for analyzing the anticancer effects of CBD in two colorectal cancer cell lines. After that, colorectal cancer cells were treated with 20 μM CBD, followed by an examination of gene expression using an mRNA-seq analysis. An mRNA-Seq analysis provides detailed and quantitative data for in-depth profiling of the transcriptome and the elucidation of various physiological and pathological conditions [41]. A differentially expressed gene (DEG) analysis showed that the expression of 1624 genes was regulated by CBD treatment in LoVo cells and that more than half of 1109 genes were downregulated by CBD. Go et al. also reported that CBD treatment downregulated 1019 genes out of 1469 regulated genes in FaDu head and neck squamous cells and 1334 out of 1019 genes in SCC15 cells, indicating that the primary function of CBD was to turn off the expression of its target genes [21]. A GO term analysis showed that CBD treatment was strongly associated with cell-proliferation-associated genes, for example, genes involved in DNA replication, cell division, DNA repair, and the cell cycle. Moreover, pathways for mitochondrial translation and mitochondrial complex I assembly were also regulated by CBD treatment, indicating that the main activity of CBD might be related to cell proliferation and mitochondrial function. This means that the expression levels of many genes are reduced by CBD treatment in cancer cells, thus contributing to its anticancer activity.

However, we found that the metallothionein family genes were the most upregulated genes in our analysis. We also confirmed that most MT family genes were upregulated by CBD treatment at both the transcriptional level and the protein level. MT genes are known to encode cysteine-rich residue proteins that can bind to several heavy metals [42]. Their biological functions are related to zinc and copper homeostasis and protection from cadmium toxicity [43]. In humans, a series of MT1 proteins are encoded by MT1 genes (MT1F, MT1G, MT1H, MT1M, and MT1X), whereas a single MT2A gene encodes the MT2 protein [44]. TCGA data showed that the expression levels of MTs were upregulated in breast cancer, nasopharyngeal cancer, ovarian cancer, bladder cancer, and melanoma, whereas MTs were downregulated in hepatocellular carcinoma and prostate cancer. MT1G expression is frequently silenced or downregulated in thyroid cancer cell lines. Restoring MT1G expression can dramatically suppress cell growth and invasiveness and induce cell cycle arrest and apoptosis by inhibiting the phosphorylation of Akt and Rb [45]. A DEG analysis using mRNA-seq has suggested that sorafenib can upregulate MT1G expression via the hypomethylation of its promoter region by binding and inhibiting DNA methyltransferase 1 (DNMT1) and increasing its promoter accessibility in hepatocellular carcinoma (HCC) cells, suggesting that MT1G might constitute a strategy for enhancing the anticancer effect of sorafenib [46]. MT1G can inhibit cell growth and the cell cycle through the PI3K/AKT signaling pathway in gastric cancer cells, as the overexpression of MT1G inhibits cell proliferation, foci formation, and cell invasion and negatively regulates p-AKT [34]. In vitro and in vivo experiments have confirmed that MT1F in colorectal cancer cells could decrease cell proliferation and colony formation, increase the rate of apoptosis to inhibit cell growth, and reduce xenograft tumor growth in MT1F-expressing mice [47]. Induced MT1H expression can inhibit cell growth by reducing colony formation and the entry of prostate cells into S and M phases in prostate cancer [48]. The overexpression of MT1H can inhibit cell proliferation and migration by suppressing the nuclear translocation of β-catenin in HepG2 and Hep3B cells [49]. Therefore, the MT family plays a critical role in tumor progression in many types of tumors.

In vivo and in silico studies have suggested that CBD can modulate the activity of the enzymes responsible for DNA methylation either by directly binding to the enzyme or indirectly modulating its activity during neurotransmitter-mediated signaling [50]. In addition, CBD can regulate the activities of methyltransferases SMYD3 and ZFP37 in human follicular cells, indicating that CBD can change the DNA methylation status [51]. The MT1G promoter is hypermethylated in colorectal tumor tissues and CRC cells [52]. Metal ions and reactive oxygen species (ROS) are known to be responsible for the expression of MTs, and such expression is regulated by metal response element-binding transcription factor-1 (MTF-1) [53]. CBD treatment can induce the expression of ROS-related genes, glutathione-deficiency-related genes, and inflammation-related genes via MTF-1 in BV-2 murine microglial cells [54]. Our current study suggests that CBD might exhibit anti-tumor activity by restoring the expression of MTs in colorectal cancer cells. Although the mechanism of MT expression in colorectal cancer cells induced by CBD has not yet been clearly identified, CBD might directly regulate MT expression via MTF-1 or indirectly regulate MT expression by downregulating DNA methylation (Figure 8). It has been known that CBD and THC have very similar chemical structures [55]. However, THC acts as an agonist on CB1 and CB2 receptors, whereas CBD exerts a negative allosteric modulatory action on CB receptors [56]. The expression levels of MTs were relatively increased by treatment with crude extract or CBD, suggesting that CBD can act on its own receptors. Therefore, MT expression might be tightly regulated in the presence of CBD.

The oral administration of CBD has a short half-life, and the oral bioavailability of CBD is approximately 6–19% in humans [57]. Clinically, the treatment of cannabinoids was tried as a form of poly-use (inhalation and ingestion, 35%), closely followed by ingestion (33%) and inhalation (30%) for symptom management in cancer patients [58]. A potential reason for the multiple methods of administration may be due to differences in the pharmacokinetics and pharmacodynamics of the cannabinoids. Liver aminotransferase enzymes have been found to be elevated in 5–20% of epilepsy patients treated with CBD in clinical trials of Epidiolex, a commercial CBD-based drug, indicating that an appropriate administration method and concentration should be established to use CBD in in vivo applications [59]. A strategy that synergistically increases the effects of CBD can be used to reduce side effects such as hepatotoxicity by reducing the treatment concentration.

Our transient transfection experiment showed a synergistic effect of the expression of MT genes on the anticancer effect of CBD. In addition, cotreatment with CBD and zinc ions showed an increase in anticancer activity compared to treatment with CBD or zinc ions alone. We assumed that the MT family expression was increased by CBD treatment and that cell death could be sequentially enhanced by elevating the cellular zinc ion concentration. The MT family can act as a chaperone for zinc ions and mediate the transport and availability of zinc to other zinc-binding proteins, suggesting that zinc ions could be involved in the synergistic effect of CBD and the MT family [60]. Synergistic zinc ions are essential for maintaining homeostasis within cells. However, they can cause cytotoxicity when their concentration within cells is excessive [61].

In conclusion, CBD requires the expression of MT family genes for its anticancer activity during cancer treatment, suggesting that CBD can affect the expression of genes involved in zinc homeostasis via MT family gene expression. These current results suggest that the regulation of zinc levels could play an important role in allowing CBD to exert enhanced anticancer effects. Although CBD treatment has shown anticancer effects on many types of cancers, its molecular mechanism remains unclear. In this study, we explained the possible interactions of MT genes, metal ions, and CBD for anticancer effects. The results of this study might provide an important clue for uncovering the mechanisms involved in the anticancer effects of CBD, such as through the regulation of intracellular homeostasis of metal ions.

## 4. Materials and Methods

### 4.1. Cell Culture and Reagents

The human colorectal cancer cells LoVo and SW480 were purchased from Korea Cell Bank (Seoul, Republic of Korea) and cultured in an RPMI-1640 (Welgene, Seoul, Republic of Korea) medium supplemented with 10% fetal bovine serum (Merck Millipore, Burlington, MA, USA) and penicillin (100 U/mL) and streptomycin (100 μg/mL) (Welgene, Seoul, Republic of Korea). These cells were cultured in a CO_2_ incubator (Thermo Fisher Scientific Inc., Waltham, MA, USA) at 37 °C with 5% CO_2_. Cannabidiol (CBD, #90080), cannabigerol (CBG, #15293), cannabichromene (CBC, #26252), and Δ^9^-tetrahydrocannabinol (Δ^9^-THC, #12068) were purchased from Cayman Chemical (Ann Arbor, MI, USA); dissolved in DMSO (Dimethyl sulfoxide); aliquoted; and stored at −20 °C. The crude extract of cherry wine hemp flower was obtained from Chuncheon Bioindustry Foundation (Chuncheon, Republic of Korea).

### 4.2. MTT Assay

Cell viability was measured using an MTT (dimethyl thiazole-2′,5′-diphenyl-2-H-tetrazolium bromide) assay (VWR Chemicals, Radnor, PA, USA). The MTT reagent was dissolved in PBS (Welgene, Seoul, Republic of Korea) at a concentration of 2.5 mg/mL to make a 5× MTT solution. Cells were seeded at 6 × 10^3^ cells/well into a 96-well cell culture plate. CBD was diluted to different concentrations (0–40 μM). Cells were then treated with CBD after washing with PBS. Absorbance was measured at 24 h intervals from 0 to 96 h after drug treatment. After removing media at the indicated time point, 200 μL of 1× MTT diluted with RPMI-1640 was added to each well, and cells were incubated at 37 °C for 4 h. After replacing 1× MTT with DMSO, absorbance was measured at a wavelength of 570 nm/690 nm using a microplate reader (Allsheng, Hangzhou, China) 30 min later. The experiment was performed in triplicate.

### 4.3. Annexin Ⅴ/PI Staining

Cells were cultured on coverslips in 60 mm cell culture plates and treated with 20 μM CBD the next day. Cell death was analyzed using an Annexin V-FITC apoptosis detection kit (BioVision, Milpitas, CA, USA). Briefly, cells were stained according to the manufacturer’s instructions. Apoptosis cells were analyzed using a FACSymphony™ A3 Cell Analyzer (BD Bioscience, Franklin Lakes, NJ, USA).

### 4.4. Flow Cytometry Analysis

Cells were cultured in 60 mm dishes and treated with CBD at 20 μM for up to 48 h. Cells and media were harvested after cells were treated with 1× trypsin in PBS. The cell pellet was collected after centrifugation at 1500 rpm for 5 min at 4 °C and fixed with 75% ethanol at 4 °C for at least 2 h. After washing twice with PBS, cells were stained with 1 mL of a staining solution containing 50 μg/mL propidium iodide and 20 μg/mL RNase A at room temperature for 30 min. Thereafter, cell cycle distribution was analyzed using a FACSymphony™ A3 Cell Analyzer (BD Bioscience, Franklin Lakes, NJ, USA) according to the manufacturer’s protocols.

### 4.5. Western Blot Analysis

Cells were cultured in 100 mm dishes and treated with drugs at 24 h after seeding cells. After drug treatment for the indicated time period, cells were harvested using scrapers and washed with PBS. Total proteins were extracted with RIPA lysis buffer (10 mM Tris-HCl pH 8.0, 1 mM EDTA, 140 mM NaCl, 1% Triton X-100, 0.1% sodium deoxycholate, 0.1% SDS) supplemented with cOmplete™ Protease Inhibitor Cocktail (Roche, Basel, Switzerland). Protein quantification was measured with Bradford reagent (Thermo Fisher Scientific, Waltham, MA, USA). Proteins (30–40 μg per sample) were then subjected to SDS-PAGE, transferred to 0.45 μm polyvinylidene fluoride (PVDF) membranes, blocked with 5% skim milk for 30 min at room temperature, and incubated with a primary antibody diluted in 1× TBST (Tris Buffered Saline containing Tween) buffer with 1% BSA at 4 °C overnight. Afterward, the membrane was washed with 1X TBST buffer and incubated at room temperature for 1 h with appropriate secondary antibodies diluted in 5% skim milk. Finally, proteins were detected using an ECL protein detection kit (GE Healthcare, Chicago, IL, USA). For MTs, metallothionein monoclonal antibody (UC1MT, 1:500) was purchased from Invitrogen (Waltham, MA, USA) and used. MT2A was purchased from My BioSource (San Diego, CA, USA, 1:1000). FLAG was purchased from Sigma-Aldrich (Burlington, MA, USA, 1:2000). β-actin antibody was purchased from Santa Cruz (Dallas, TX, USA, 1:1000).

### 4.6. mRNA-seq Analysis

Cells were cultured in 100 mm cell culture dishes for 24 h and then treated with 20 μM CBD. Total RNA was isolated using TRIzol Reagent (Invitrogen, Waltham, MA, USA). RNA quality was assessed with an Agilent 2100 bioanalyzer (Agilent Technologies, Amstelveen, The Netherlands). RNA quantification was performed using an ND-2000 Spectrophotometer (Thermo Fisher Scientific, Waltham, MA, USA). Libraries were prepared from total RNA using a NEBNext Ultra II Directional RNA-Seq Kit (NEB, Ipswich, MA, USA). The isolation of mRNA was performed using a Poly(A) RNA Selection Kit (LEXOGEN, Vienna, Austria). Isolated mRNAs were used for cDNA synthesis followed by shearing according to the manufacturer’s instructions. Indexing was performed using Illumina indexes 1–12. The enrichment step was carried out using PCR. Subsequently, libraries were checked using a TapeStation HS D1000 Screen Tape (Agilent Technologies, Amstelveen, The Netherlands) to evaluate the mean fragment size. Quantification was performed using a library quantification kit using a StepOne Real-Time PCR System (Life Technologies, Carlsbad, CA, USA). High-throughput sequencing was performed as paired-end 100 sequencing using a NovaSeq 6000 (Illumina, San Diego, CA, USA).

### 4.7. Sequencing Data Analysis

Quality control of raw sequencing data was performed using FastQC v0.11.9 (http://www.bioinformatics.babraham.ac.uk/projects/fastqc/, accessed on 26 October 2023). Adapter and low-quality reads (<Q20) were removed using FASTX_Trimmer v0.0.14 (http://hannonlab.cshl.edu/fastx_toolkit/, accessed on 26 October 2023) and BBMap v35.74 (https://sourceforge.net/projects/bbmap/, accessed on 26 October 2023). Trimmed reads were then aligned to the human genome (ENSEMBL-grch38release 91) using TopHat v2.1.1 [62]. Read count (RC) data were processed based on the Fragments Per kb per Million reads (FPKM) + geometric normalization method using Cufflinks v2.2.1 [63]. Data mining and graphic visualization were performed using ExDEGA v3.2.1 (Ebiogen, Seoul, Republic of Korea). A gene ontology enrichment analysis was performed using DAVID Bioinformatics Resources 6.7 r4 (https://david.ncifcrf.gov/, accessed on 22 June 2022).

### 4.8. TCGA Data Analysis

Two TCGA databases, UALCAN (http://ualcan.path.uab.edu/, accessed on 8 December 2022) and TNMplot (https://tnmplot.com/analysis/, accessed on 16 October 2023), were used to analyze the mRNA and protein levels of the MT family in cancers [64,65].

### 4.9. qPT-PCR Analysis

Cells were cultured in 100 mm cell culture dishes for 24 h and then treated with 20 μM CBD. Total RNA was isolated using TRIzol Reagent. cDNA was synthesized using 2 µg of total RNA, oligo dT (Promega, Madison, WI, USA), and M-MLV reverse transcriptase (Promega, Madison, WI, USA). The primers used for quantitative PCR are shown in Appendix A. qPCR reactions were performed in the presence of SYBR Green dye in TOPreal™ qPCR 2X PreMIX (Enzynomics, Daejeon, Republic of Korea). The relative expression levels were calculated using the 2^−ΔΔCt^ method [66]. Fold change (FC) values ≥ 2 were considered to be statistically significant. The CT (threshold cycle) values of target genes were normalized to the CT values of β-actin housekeeping genes.

### 4.10. Immunofluorescence

Cells were cultured on coverslips in 6-well plates and treated with drugs at 24 h after seeding cells. They were then fixed with 4% paraformaldehyde solution for 10 min. Cells were permeabilized with 0.1% Triton X-100 diluted in PBS for 3 min, blocked with 5% skim milk in PBS at room temperature for 1.5 h, and incubated with a diluted primary antibody (1:200) in 5% skim milk at room temperature for 2 h. After washing with PBS, cells were incubated with secondary Alexa 488 goat anti-mouse IgG antibodies (1:200; Abcam, Boston, MA, USA) in 5% skim milk at room temperature for 1 h. Counterstaining was performed with DAPI (1:2000) in PBS. Cells were observed under a confocal microscope (Nikon, Tokyo, Japan) for immunofluorescence images.

### 4.11. Transient Transfection

The expression vectors of MT1G and MT2A were purchased from Origene (RC204741 and RC202748, Rockville, MD, USA). About 60% confluent cells in 60 mm cell culture dishes were used for transfection with Lipofectamine 2000 (Invitrogen, Waltham, MA, USA). Cells were transfected with 3 μg of pCMV6-empty vector or MT overexpression vector using 5 μL of Lipofectamine in a serum-free medium and incubated for 12 h. The medium was then replaced with an FBS-containing medium, and cells were treated with CBD after 12 h.

### 4.12. Statistical Analysis

A statistical analysis was performed using GraphPad Prism 8.0 and ImageJ v1.54d. Graphs are presented as mean ± standard deviation (SD) from at least 3 to 4 independent experiments. Groups of two were analyzed with Student’s *t*-test. Statistical significance was considered when the *p*-value was less than 0.05, and it is indicated by asterisks (* *p* ≤ 0.05; ** *p* ≤ 0.01).

## Figures and Tables

**Figure 1 ijms-24-16621-f001:**
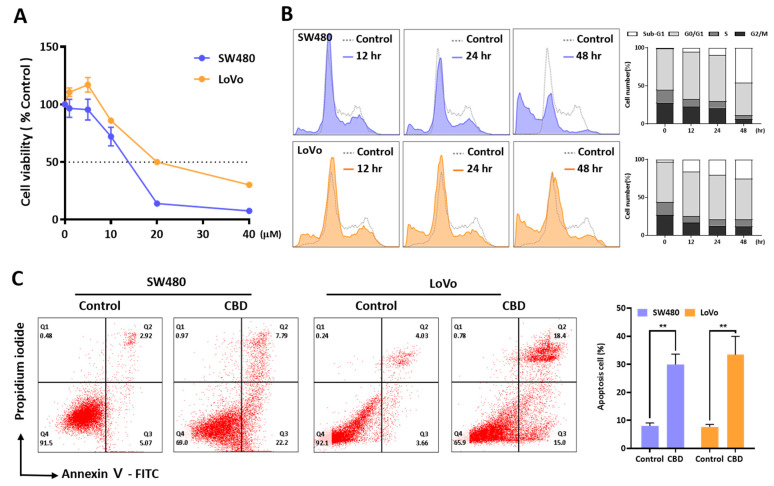
Growth inhibition effect and anticancer activity of CBD in colorectal cancer cells. (**A**) Cell viability was determined using MTT assay. CBD inhibits growth of two colorectal cancer cell lines in a dose-dependent manner. The dotted line means cell viability 50%. IC_50_ was determined as 12.86 ± 0.527 μM for SW480 cells and 23.03 ± 4.09 μM for LoVo cells. (**B**) Flow cytometry analysis was performed to examine changes in cell cycle distribution caused by CBD. (**C**) Apoptosis cells were calculated after staining with Annexin V/PI-FITC using flow cytometry. Dead cell population was increased after CBD treatment for 24 h. (**, *p* ≤ 0.01).

**Figure 2 ijms-24-16621-f002:**
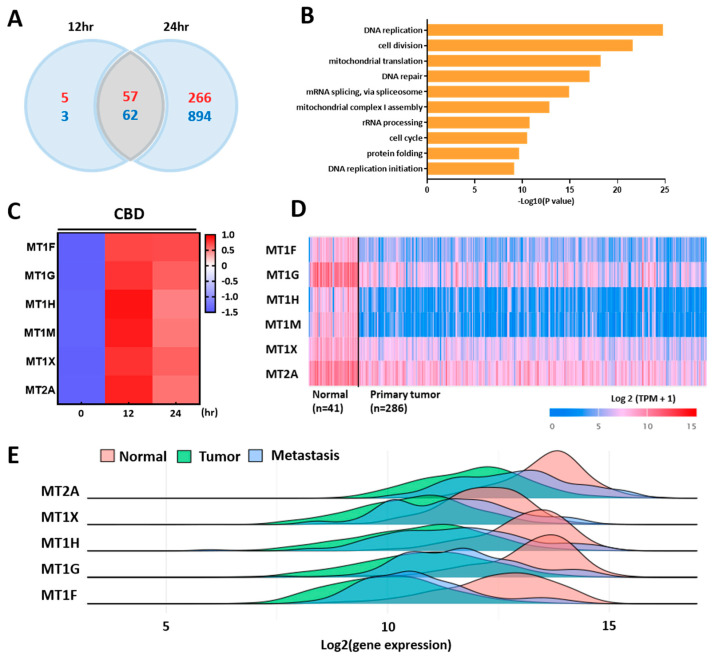
mRNA-seq analysis of gene expression profiles following CBD treatment. (**A**) mRNA-seq analysis was performed after 12 h or 24 h treatment of CBD. A number of genes were changed by CBD treatment at each time point. (red; upregulated genes, blue; downregulated genes) (**B**) GO analysis showed that CBD mainly regulated the expression of genes associated with DNA replication or cell division. (**C**) Heat map showing that the expression of the metallothionein gene family was increased by CBD in a time-dependent manner. (**D**) TCGA analysis suggesting that expression levels of most MT family genes were decreased in primary tumor tissues compared to those in normal tissues. (**E**) Expression levels of most MT genes were significantly decreased during tumorigenesis and metastasis of colorectal cancers.

**Figure 3 ijms-24-16621-f003:**
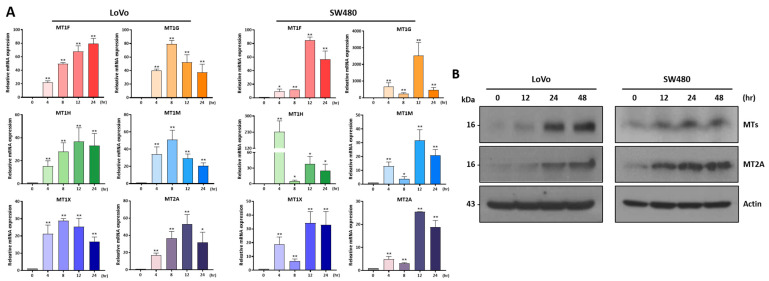
qPCR analysis of metallothionein gene family during CBD treatment. (**A**) qRT-PCR was performed to examine expression level changes in metallothionein gene family by CBD treatment in two colorectal cancer cell lines. MT family genes were upregulated by CBD treatment in a time-dependent manner (* *p* ≤ 0.05; ** *p* ≤ 0.01). (**B**) Western blot analysis was performed for 20 μM CBD-treated colorectal cancer cells. Expression levels of MTs and MT2A were increased by CBD treatment in a time-dependent manner.

**Figure 4 ijms-24-16621-f004:**
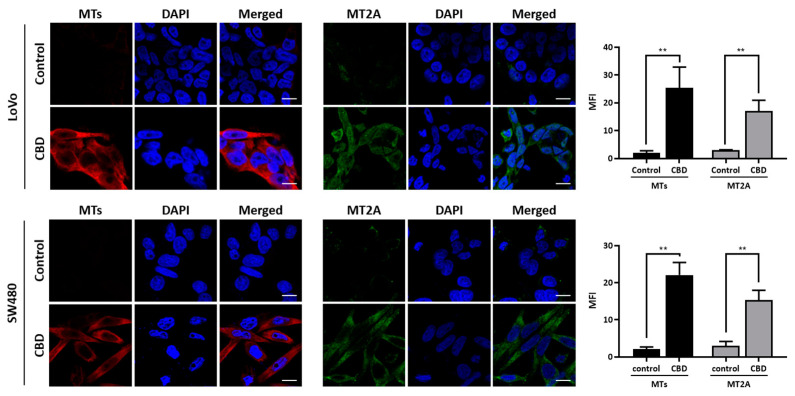
Immunostaining analysis of metallothionein proteins after CBD treatment. Cells were treated with 20 μM CBD for 48 h, stained with MT antibody (red), MT2A antibody (green) and DAPI (blue). MT and MT2A levels were significantly increased in the cytoplasmic region after CBD treatment. (**, *p* ≤ 0.01). Scale bar, 10 µm.

**Figure 5 ijms-24-16621-f005:**
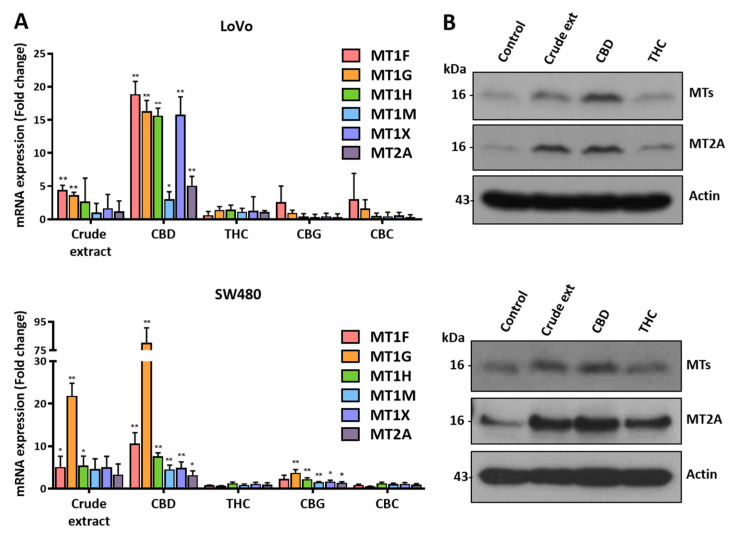
Expression levels of MT family genes after treatment with various cannabinoids. (**A**) Cells were treated with five kinds of cannabinoids, including crude extract, followed by qRT-PCR analysis. The expression of MT mRNA was increased by treatment with crude extract or CBD (* *p* ≤ 0.05; ** *p* ≤ 0.01). (**B**) Western blot analysis showed that MT protein expression was also increased by crude extract or CBD.

**Figure 6 ijms-24-16621-f006:**
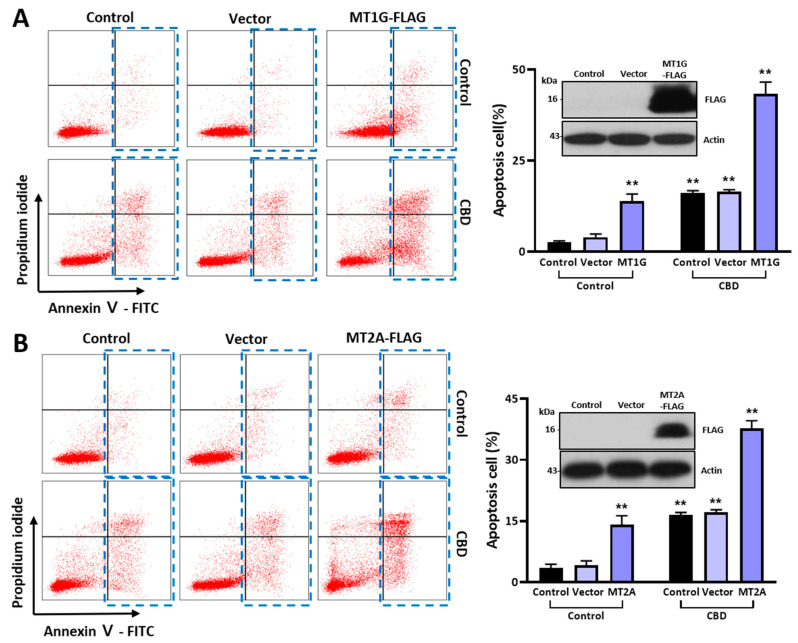
Synergistic effect of MT overexpression on anticancer effect of CBD. (**A**) SW480 cells were transfected with MT1G overexpression vector and treated with 20 μM CBD for 48 h. Apoptosis cells were analyzed using flow cytometry after staining with Annexin V/PI-FITC. (**B**) SW480 cells were transfected with MT2A overexpression vector and treated with 20 μM CBD for 48 h. Apoptosis cells were analyzed using flow cytometry after staining with Annexin V/PI-FITC. In both transient transfection experiments, the dead cell population (blue rectangles) was increased in the group with overexpression of MTs and CBD treatment. (** *p* ≤ 0.01).

**Figure 7 ijms-24-16621-f007:**
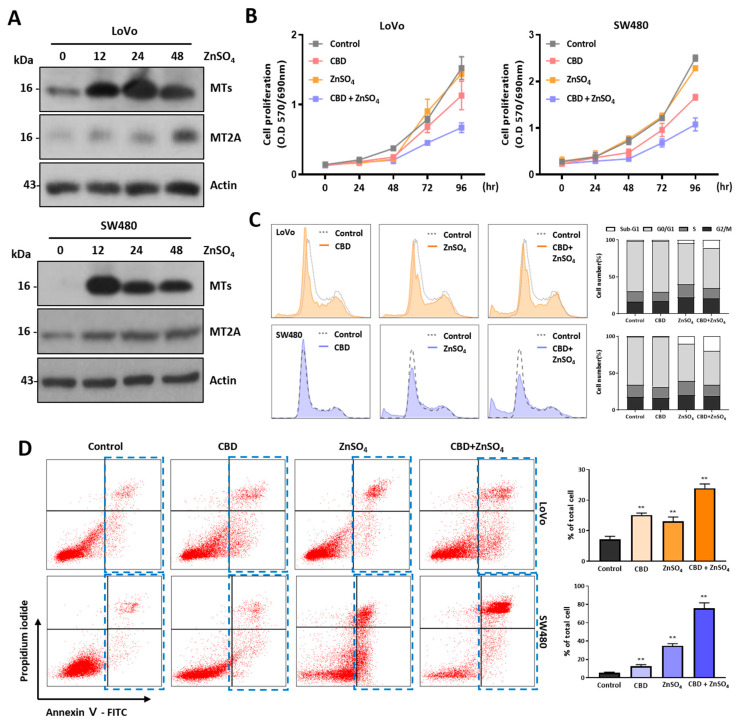
Effect of combination treatment of CBD and zinc ions on anticancer activity. (**A**) Cells were treated with zinc ions for different time periods. MT and MT2A proteins were increased by zinc ion treatment in two colorectal cancer cell lines. (**B**) Cell viability was examined after combination treatment with sublethal concentration of CBD and zinc ions using MTT assay. Cell viability was significantly decreased by combination treatment with 5 μM CBD and zinc ions. (**C**) Flow cytometry analysis showed that the dead cell population was increased by combination treatment with 5 μM CBD and zinc ions. (**D**) Apoptosis cells were analyzed using flow cytometry after Annexin V/PI dual staining following combination treatment with 5 μM CBD and zinc ions. The dead cell population (blue rectangles) was increased after combination treatment with CBD and zinc ions (** *p* ≤ 0.01).

**Figure 8 ijms-24-16621-f008:**
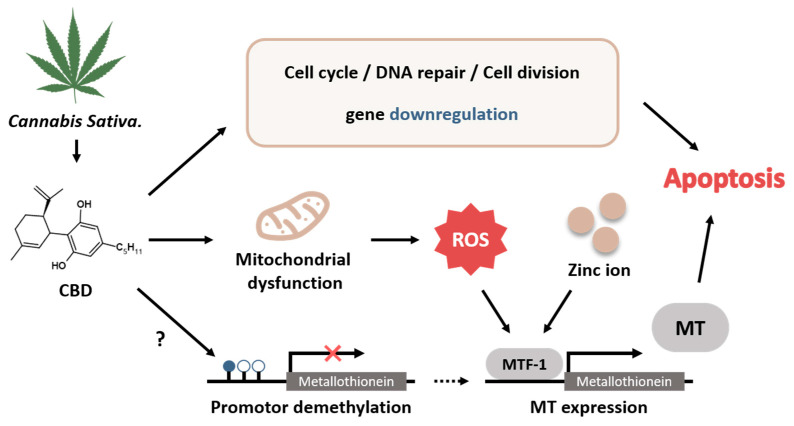
Graphic summary of MT-mediated anticancer effects of CBD and related signaling pathways. CBD, cannabidiol; ROS, reactive oxygen species; MTF-1, metal response element-binding transcription factor-1; MT, metallothionein.

## Data Availability

All data generated or analyzed during this study are included in this article. Further inquiries can be directed to the senior corresponding author.

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
