# Peer review of "Metallothionein Family Proteins as Regulators of Zinc Ions Synergistically Enhance the Anticancer Effect of Cannabidiol in Human Colorectal Cancer Cells"

_ijms, 2023, doi:10.3390/ijms242316621_

Round 1
Reviewer 1 Report
Comments and Suggestions for Authors
The study by I.-S. Kwon et al. investigated the effects of cannabidiol (CBD) and related compounds as anticancer candidates. The authors performed a series of well designed experiments to demonstrate the efficacy of CBD. Among the most intriguing results is the differential regulation of genes in response to CBD, in particular, a strong activation of metallothionein family mRNAs. Also, the combinations of CBD at non-toxic concentrations with zinc yielded a sizeable synergy in cell death.
Addressing a few questions would further strengthen the study:
1. What is the mechanism of coordinated up-regulation of all tested MT family genes by CBD? Which signaling pathways and transcription factors might be involved?
2. Why CBD is potent whereas other cannabinoids are not (Figure 5)? What is so special about CBD structure and properties?
3. Please show statistical errors on bar diagrams (Figures 6, 7).
4. A scheme summarizing the mechanistic links (end of Discussion) is worthy.
5. What is the opinion of the authors in regard to therapeutic perspective of CBD+MT combinations?
Comments on the Quality of English LanguageIn general, the quality of the language is decent although the text would definitely win from editing. Errors are scientific, not as much grammatical. For instance, line 226: we first determined that 20 μM was the optimal concentration for the anti-cancer effect...- what do you mean optimal? Please explain the choice of this concentration. Line 245: MT genes are cysteine-rich residue proteins... - should be MT genes encode the proteins.
Author Response
Reviewer 1
The study by I.-S. Kwon et al. investigated the effects of cannabidiol (CBD) and related compounds as anticancer candidates. The authors performed a series of well designed experiments to demonstrate the efficacy of CBD. Among the most intriguing results is the differential regulation of genes in response to CBD, in particular, a strong activation of metallothionein family mRNAs. Also, the combinations of CBD at non-toxic concentrations with zinc yielded a sizeable synergy in cell death.
Addressing a few questions would further strengthen the study:
- What is the mechanism of coordinated up-regulation of all tested MT family genes by CBD? Which signaling pathways and transcription factors might be involved?
Answer) As you recommended, we described the plausible mechanism on how CBD might induce MT expression by adding several references in the Discussion section.
- Why CBD is potent whereas other cannabinoids are not (Figure 5)? What is so special about CBD structure and properties?
Answer) We thought that MT expression was differently regulated by receptor preference on cannabinoids. As you recommended, we described explanation on differences of receptor preference between CBD and other cannabinoids in the discussion section.
- Please show statistical errors on bar diagrams (Figures 6, 7).
Answer) As you suggested, we performed a statistical analysis on the data, and modified the graphs in Figure 6 and 7.
- A scheme summarizing the mechanistic links (end of Discussion) is worthy.
Answer) As recommended, a graphical summary was added in Figure 8.
- What is the opinion of the authors in regard to therapeutic perspective of CBD+MT combinations?
Answer) In this study, we introduced possible interaction for anti-cancer effect via CBD, MT gene, and metal ion. These studies will provide important clue in uncovering the mechanism of anticancer effect by CBD through regulation of intracellular homeostasis of metal ion.
In general, the quality of the language is decent although the text would definitely win from editing. Errors are scientific, not as much grammatical. For instance, line 226: we first determined that 20 μM was the optimal concentration for the anti-cancer effect...- what do you mean optimal? Please explain the choice of this concentration. Line 245: MT genes are cysteine-rich residue proteins... - should be MT genes encode the proteins.
Answer) As you recommended, we corrected the sentences in the text.

Reviewer 2 Report
Comments and Suggestions for Authors
The possible use of CBD, the main non-psychotropic component of cannabis sativa, as an anti-cancer agent is a critical hot topic of cancer research that engages the scientific community in preclinical studies aimed at understanding its mechanism of action, safety, and efficacy.
In this contest, the authors proposed to analyze the action mechanism of CBD, and cherry wine hemp flower extract in human colorectal cancer cells.
The aim of the research is interesting and the rationale of the work is clear, however, some criticism of the work should be addressed before the endorsement of the publication.
RESULTS
1)
It is very important that the authors verify that the significant drop observed in cell viability at high doses of CBD (20-40µM) (Figure1A) is not due to the loss of detached cells or to the formation of aggregates of CBD that precipitate on the cells inducing cell death.
Indeed, it has been demonstrated that CBD affects the cell adhesion properties of some cancer cells, by favouring cell detachment. As reported in the material and methods section MTT assay was performed by removing the medium from the plates before the addition of the formazan reagent. With this procedure, all cells eventually detached upon the CBD treatment could be lost, thus affecting the measurements of the cell viability which should be underestimated. Therefore, I suggest that the authors try (1) to perform the MTT assay by collecting the media containing the possibly detached cells and suspending the cell pellet in the reagent before it is added to the corresponding well, or (2) to directly add the reagent in culture medium at the end of CBD treatment.
Furthermore, since CBD (that is a hydrophobic compound) tends to aggregate in water solutions (i.e. medium) at high concentrations, the authors should verify the eventual formation of aggregates in the medium containing 20-40µM CBD. The author could perform an MTT assay on cells treated with a medium containing 20-40µM CBD filtered on 0.22µm filters.
2)
Some experimental results reported in the manuscript were not subjected to quantitative analysis or statistical analysis. The legend of each figure has to clearly report the details of the experimental condition analyzed, including drug concentrations, the timing of treatments, the number of biological replicates, the statistical test applied, and the statistical significance of the data, in order to make more intelligible the results reported.
In Figure 1B, the FACS data should be quantitatively analyzed with the opportune software in order to determine the % of cells in different phases of the cell cycle, including the sub-G1 phase that correlates with DNA fragmentation and apoptosis. Qualitatively is more appreciable an enrichment in the sub-G1 phase than the G1 arrest after CBD treatment! The FACS data reported in Figure 7C should also be quantitatively analyzed as above reported.
Why do the graph bars of Figure 1C and 7D not contain standard deviation? How many biological replicates have been performed? The graph has to report standard deviation and statistical analysis.
In Figure 3B, in Figure 5B and in Figure 7A a representative western blot of each experiment was reported. It is important that a densitometric analysis was done for each western blot performed (at least three biological replicates) and that the mean ± standard deviation is plotted in a graph to be inserted in the same Figures. Notably, the referee can ask the authors to see the results of the different western blot experiments performed.
For the experiment results reported in Figure 7B the authors have to add information about the concentration of ZnSO4 used.
MATERIAL AND METHODS
The material and methods section has to be extended by adding information concerning the timing of the treatment (par 4.3, 4.6 and 4.9) and the dilution of the primary antibodies (par 4.5)
DISCUSSION
Focusing on published in vitro studies, the effect of CBD on cancer cell viability ranges from no effect to a modest reduction, and to significant cytotoxicity depending on concentrations, cancer cell lines, cell growth conditions, the performed assays, and the time of CBD exposure. (Fowler CJ. Delta(9) -tetrahydrocannabinol and cannabidiol as potential curative agents for cancer: A critical examination of the preclinical literature. Clin Pharmacol Ther. 2015 Jun;97(6):587-96. doi: 10.1002/cpt.84. Epub 2015 May 2. PMID: 25669486. ------- Pagano S, Coniglio M, Valenti C, Federici MI, Lombardo G, Cianetti S, Marinucci L. Biological effects of Cannabidiol on normal human healthy cell populations: Systematic review of the literature. Biomed Pharmacother. 2020 Dec;132:110728. doi: 10.1016/j.biopha.2020.110728. Epub 2020 Oct 7. PMID: 33038581. -------Valenti C, Billi M, Pancrazi GL, Calabria E, Armogida NG, Tortora G, Pagano S, Barnaba P, Marinucci L. Biological effects of cannabidiol on human cancer cells: Systematic review of the literature. Pharmacol Res. 2022 Jul;181:106267. doi: 10.1016/j.phrs.2022.106267. Epub 2022 May 25. PMID: 35643249. -------and other more recent review). At the basis of this variability, at least in part, there is the poor solubility of CBD in aqueous solutions, such as the culture medium, that causes the formation of aggregates and precipitates in solutions at high concentrations, which is influenced by the serum concentration. Furthermore, growth factors present in the serum supplemented to the culture medium can modulate CBD effects on cells in vitro by directly interfering with the interaction between CDB and specific growth factor receptors (Sainz-Cort A, Müller-Sánchez C, Espel E. Anti-proliferative and cytotoxic effect of cannabidiol on human cancer cell lines in presence of serum. BMC Res Notes. 2020 Aug 20;13(1):389. doi: 10.1186/s13104-020-05229-5. PMID: 32819436; PMCID: PMC7441616.------D'Aloia A, Ceriani M, Tisi R, Stucchi S, Sacco E, Costa B. Cannabidiol Antiproliferative Effect in Triple-Negative Breast Cancer MDA-MB-231 Cells Is Modulated by Its Physical State and by IGF-1. Int J Mol Sci. 2022 Jun 27;23(13):7145. doi: 10.3390/ijms23137145. PMID: 35806150; PMCID: PMC9266539.)
For this reason, it is important that the authors discuss in their manuscript these issues impacting the anti-proliferative/ cytotoxic properties of CBD in vitro referring to the literature data.
In the discussion section, the authors should also critically discuss the translatability of their study in vivo on cancer patients. In this regard, based on literature data, the authors should discuss whether the concentrations of CBD used in their study are achievable in vivo and with what type of administration. These data are important in the perspective of the use of CBD in cancer patients, to deduce its potential SAFETY and anti-cancer effectiveness.
Author Response
Reviewer 2
The possible use of CBD, the main non-psychotropic component of cannabis sativa, as an anti-cancer agent is a critical hot topic of cancer research that engages the scientific community in preclinical studies aimed at understanding its mechanism of action, safety, and efficacy.
In this contest, the authors proposed to analyze the action mechanism of CBD, and cherry wine hemp flower extract in human colorectal cancer cells.
The aim of the research is interesting and the rationale of the work is clear, however, some criticism of the work should be addressed before the endorsement of the publication.
RESULTS
1)
It is very important that the authors verify that the significant drop observed in cell viability at high doses of CBD (20-40µM) (Figure1A) is not due to the loss of detached cells or to the formation of aggregates of CBD that precipitate on the cells inducing cell death.
Indeed, it has been demonstrated that CBD affects the cell adhesion properties of some cancer cells, by favouring cell detachment. As reported in the material and methods section MTT assay was performed by removing the medium from the plates before the addition of the formazan reagent. With this procedure, all cells eventually detached upon the CBD treatment could be lost, thus affecting the measurements of the cell viability which should be underestimated. Therefore, I suggest that the authors try (1) to perform the MTT assay by collecting the media containing the possibly detached cells and suspending the cell pellet in the reagent before it is added to the corresponding well, or (2) to directly add the reagent in culture medium at the end of CBD treatment.
Answer) Thanks for your valuable suggestion. We performed the MTT assay to check changes in cell proliferation by CBD treatment. We also considered lose of buoyant dead cells as you point out. That’why we also performed FACS experiment and annexin V experiment. For those experiment, we harvested cell samples including buoyant dead cells (supernant) and the attached cells with scraper as you suggested. Therefore, these three separate experiments suggest that 20 uM CBD has anticancer activity in colorectal cancer cells. However, we will perform MTT analysis to get fine data as your suggestion in the future experiment.
Furthermore, since CBD (that is a hydrophobic compound) tends to aggregate in water solutions (i.e. medium) at high concentrations, the authors should verify the eventual formation of aggregates in the medium containing 20-40µM CBD. The author could perform an MTT assay on cells treated with a medium containing 20-40µM CBD filtered on 0.22µm filters.
Answer) DMSO have been used as solvents for hydrophobic compounds such as CBD. As CBD is a hydrophobic compound, and we also dissolved it in DMSO. Many other papers also described that CBD was dissolved in DMSO and treated in cell culture media to analyze molecular effects. (Fei Wang et al. Cannabidiol-induced crosstalk of apoptosis and macroautophagy in colorectal cancer cells involves p53 and Hsp70. Cell Death Discov. 2023 ; Huang T et al. Cannabidiol inhibits human glioma by induction of lethal mitophagy through activating TRPV4. Autophagy. 2021 ; Viereckl MJ et al. Cannabidiol and Cannabigerol Inhibit Cholangiocarcinoma Growth In Vitro via Divergent Cell Death Pathways. Biomolecules. 2022) Filtering method is also good way for drug treatment, but we concerned to reduce the effectiveness of CBD in this case.
To avoid your concern, our lab. have also established drug treatment method in the presence of serum according to our previous experiences. That’s why we adopted CBD (in DMSO) treatment method after mixing and vortexing with culture media. We think that the filtering method will be helpful for our future experiment. Thank you for your valuable comments.
2)
Some experimental results reported in the manuscript were not subjected to quantitative analysis or statistical analysis. The legend of each figure has to clearly report the details of the experimental condition analyzed, including drug concentrations, the timing of treatments, the number of biological replicates, the statistical test applied, and the statistical significance of the data, in order to make more intelligible the results reported.
In Figure 1B, the FACS data should be quantitatively analyzed with the opportune software in order to determine the % of cells in different phases of the cell cycle, including the sub-G1 phase that correlates with DNA fragmentation and apoptosis. Qualitatively is more appreciable an enrichment in the sub-G1 phase than the G1 arrest after CBD treatment! The FACS data reported in Figure 7C should also be quantitatively analyzed as above reported.
Answer) As following your suggestion, we performed quantitative analysis on FACS experiment with Flowjo software and added graphs. As pointed out, the increase of the sub-G1 phase cells tends to be more noticeable, so we described this explanation in the result section.
Why do the graph bars of Figure 1C and 7D not contain standard deviation? How many biological replicates have been performed? The graph has to report standard deviation and statistical analysis.
Answer) We performed most of experiments at least three times. For Figure 1C and 7D, we performed a statistical analysis on data, and added the graphs containing standard deviation.
In Figure 3B, in Figure 5B and in Figure 7A a representative western blot of each experiment was reported. It is important that a densitometric analysis was done for each western blot performed (at least three biological replicates) and that the mean ± standard deviation is plotted in a graph to be inserted in the same Figures. Notably, the referee can ask the authors to see the results of the different western blot experiments performed.
Answer) Thank you for your comments. We performed densitometric analysis on Western blot data using ImageJ program, and added the graphs on densitometric analysis in Supplementary Figure 1, 2, 3.
Here, we added some different Western blot data you asked.
different Western blot of Fig3
different Western blot of Fig 5
different Western blot of Fig 7
For the experiment results reported in Figure 7B the authors have to add information about the concentration of ZnSO4 used.
Answer) As suggested, the treatment concentration of ZnSO4 was described in result 2.6.
MATERIAL AND METHODS
The material and methods section has to be extended by adding information concerning the timing of the treatment (par 4.3, 4.6 and 4.9) and the dilution of the primary antibodies (par 4.5)
Answer) As you suggested, we described the drug treatment times in 4.3, 4.6, and 4.9 of Materials and Methods, and revised 4.5 in more detail.
DISCUSSION
Focusing on published in vitro studies, the effect of CBD on cancer cell viability ranges from no effect to a modest reduction, and to significant cytotoxicity depending on concentrations, cancer cell lines, cell growth conditions, the performed assays, and the time of CBD exposure. (Fowler CJ. Delta(9) -tetrahydrocannabinol and cannabidiol as potential curative agents for cancer: A critical examination of the preclinical literature. Clin Pharmacol Ther. 2015 Jun;97(6):587-96. doi: 10.1002/cpt.84. Epub 2015 May 2. PMID: 25669486. ------- Pagano S, Coniglio M, Valenti C, Federici MI, Lombardo G, Cianetti S, Marinucci L. Biological effects of Cannabidiol on normal human healthy cell populations: Systematic review of the literature. Biomed Pharmacother. 2020 Dec;132:110728. doi: 10.1016/j.biopha.2020.110728. Epub 2020 Oct 7. PMID: 33038581. -------Valenti C, Billi M, Pancrazi GL, Calabria E, Armogida NG, Tortora G, Pagano S, Barnaba P, Marinucci L. Biological effects of cannabidiol on human cancer cells: Systematic review of the literature. Pharmacol Res. 2022 Jul;181:106267. doi: 10.1016/j.phrs.2022.106267. Epub 2022 May 25. PMID: 35643249. -------and other more recent review). At the basis of this variability, at least in part, there is the poor solubility of CBD in aqueous solutions, such as the culture medium, that causes the formation of aggregates and precipitates in solutions at high concentrations, which is influenced by the serum concentration. Furthermore, growth factors present in the serum supplemented to the culture medium can modulate CBD effects on cells in vitro by directly interfering with the interaction between CDB and specific growth factor receptors (Sainz-Cort A, Müller-Sánchez C, Espel E. Anti-proliferative and cytotoxic effect of cannabidiol on human cancer cell lines in presence of serum. BMC Res Notes. 2020 Aug 20;13(1):389. doi: 10.1186/s13104-020-05229-5. PMID: 32819436; PMCID: PMC7441616.------D'Aloia A, Ceriani M, Tisi R, Stucchi S, Sacco E, Costa B. Cannabidiol Antiproliferative Effect in Triple-Negative Breast Cancer MDA-MB-231 Cells Is Modulated by Its Physical State and by IGF-1. Int J Mol Sci. 2022 Jun 27;23(13):7145. doi: 10.3390/ijms23137145. PMID: 35806150; PMCID: PMC9266539.)
For this reason, it is important that the authors discuss in their manuscript these issues impacting the anti-proliferative/ cytotoxic properties of CBD in vitro referring to the literature data.
In the discussion section, the authors should also critically discuss the translatability of their study in vivo on cancer patients. In this regard, based on literature data, the authors should discuss whether the concentrations of CBD used in their study are achievable in vivo and with what type of administration. These data are important in the perspective of the use of CBD in cancer patients, to deduce its potential SAFETY and anti-cancer effectiveness.
Answer) As you suggested, we described the explanation in citing previous papers on anticancer effects the CBD, translatability on cancer patients, and issues on CBD concentration. We could describe our further enriched discussion for your valuable comments. Thanks again.

Round 2
Reviewer 2 Report
Comments and Suggestions for Authors
The authors have addressed all of my comments, including those concerning the missing quantitative analysis.
The manuscript is now suitable for publication.